# OpenReview forum: "Neural Interactive Proofs"
_ICLR.cc/2025/Conference — ICLR 2025 Poster_

### Official Review · Reviewer_AhRe · 2024-10-27

**Soundness:** 2
**Presentation:** 2
**Contribution:** 2
**Rating:** 5
**Confidence:** 4

**Summary:**

The paper considers a prover-verifier game to improve the correctness of the output of ML models. The verifier, which cannot direct prove the correctness, engages with a set of provers whose outputs are not trustable. The interaction between verifier and provers can make the final answer more trustable. It also formulates their interactions as a zero-knowledge proof.

Simple experiments are conducted on two use cases: testing graph isomorphism and code validation.

**Strengths:**

The formulation of the problem into a zero-knowledge based verifier-prover interaction.

**Weaknesses:**

While I like the formulation, I have some concerns about this paper:

1. It is unclear on the zero-knowledge aspect. Why do we need to have the zero-knowledge in their interactions? The argument on the potential model stealing needs to be more carefully consolidated.

2. The actual technical contribution is limited. Neural interactive proof is technically explained with worst-case loss and Stackelberg game. First, I wasn't able to understand the paragraph under Proposition 2, due to the confusing and undefined notations. This seriously affects my understanding about the worst case loss. Second, the bi-level optimisation used to solve the Stackelberg game is obvious, and for a verifier-prover game, I don't think we need to go through a definition of Stackelberg game to reach the bi-level optimisation.

3. Many technical developments in the paper are not actually implemented and experimented, including the zero-knowledge interactions. This, combined with many typos in the paper (e.g., "a node a similar node the second graph"), shows that the paper requires more developments before it becomes publishable.

4. The experiments were conducted on two simple experiments. It is unclear if the method is actually better and can generalise to more complex settings.

In summary, I like the formulation of the problem as a zero-knowledge based verifier-prover game, but such reduction is not well motivated and the actual technical contribution (e.g., zero-knowledge) is not consolidated, neither theoretically nor practically.

**Questions:**

Please see the weaknesses, and I would like to see more discussions, and technical developments, related to the zero-knowledge proofs.

---

> ### Author Response · Authors · 2024-11-21
> **Reply to Reviewer AhRe (Part 1)**
>
> We thank the reviewer for their careful reading and commentary on our paper. In what follows, we address the weaknesses and questions raised by the reviewer, in light of our updates to the paper described in the official comment further above. In particular, we have expanded our empirical results and linked them more closely to our theoretical contributions (the latter of which we also explain and justify further below). We also seek to address what we view to be some misunderstandings of our work in the reviewer’s commentary. To the extent that these replies address the reviewer’s concerns, we kindly request that they consider the impact this might have on their assigned scores; to the extent that they fail to address the reviewer’s concerns, we warmly welcome any further questions or comments.
>
> ## Weaknesses
>
> **“It is unclear on the zero-knowledge aspect”**
>
> We agree with the reviewer that the original version of the paper was too brief in motivating our discussion of zero-knowledge protocols. The idea here is that while prover-verifier games may describe a training setup (in today’s current ML paradigm where there is a training-deployment dichotomy), in the future we will likely have large numbers of AI systems and services interacting with one another in order to solve tasks. While we may want such systems to be able to query one another, we may not wish for agents to gain additional knowledge from doing so (perhaps because it represents private information, or could imbue the agent with new, potentially dangerous capabilities).
>
> While this risk is not novel, the concept of zero-knowledge interactions between such agents provides a firm theoretical foundation for addressing such problems. On the other hand (from the verifier’s perspective instead of the prover’s), it also suggests a fundamental limit to the amount that might be learnt from interacting with another, more powerful agent. Thus, while using zero-knowledge protocols is certainly not “necessary”, it may provide additional information security. To support this hypothesis, we have now also included some preliminary experimental results validating our zero-knowledge training scheme.
>
> **Worst-Case Optimisation and Stackelberg Equilibria**
>
> Regarding the reviewer’s confusion about the paragraph under Proposition 2, we kindly request that they point out the specific notation that is unclear, so that we correct or further explain this if necessary. Regarding the use of bi-level optimisation to solve the Stackelberg game, we believe the reviewer may have misunderstood the motivation behind this approach. We do not introduce the idea of a Stackelberg equilibrium in order to motivate bi-level optimisation; rather, it is Stackelberg equilibria of the prover-verifier games that correspond to valid proof systems (see Theorems 1-3), and seeking the latter in turn motivates using bi-level optimisation. Thus, the idea of Stackelberg equilibria is in fact crucial to the correspondence between prover-verifier games and proof protocols, whereas bi-level optimisation is not.
>
> **Unimplemented developments**
>
> We agree with the reviewer that a weakness of the original version of our paper was that we did not include experimental results for the zero-knowledge protocols and some of the other protocols in the paper. This shortcoming has now been rectified, and the additional results are described further in the update summary above. We have also implemented methods for adversarial training and algorithms for Stackelberg Policy Gradient [(Fiez et al., 2020)](https://proceedings.mlr.press/v119/fiez20a.html), Learning with Opponent-Learning Awareness [(Foerster et al., 2018)](https://arxiv.org/abs/1709.04326), and Stable Opponent Shaping [(Letcher et al., 2019)](https://arxiv.org/abs/1811.08469) in our codebase.
>
> **Typos**
>
> We respectfully disagree with the reviewer that the paper has “many” typos, or that the existence of extremely minor typos (such as "a node a similar node the second graph") means that the paper requires significant development before it can be published. We will of course make sure that we rigorously check for any remaining small typos before the full paper is published, and if the reviewer has identified any more typos beyond the single instance they included in their original review, we kindly request that they make these known to us so that they can be fixed.

---

> ### Author Response · Authors · 2024-11-21
> **Reply to Reviewer AhRe (Part 2)**
>
> **“Simple experiments”**
>
> While the reviewer is correct that the Graph Isomorphism experiments are deliberately simple (as this allows us benchmark our empirical results against theoretical bounds, and to more easily assess the influence of key variables such as the network size and amount of training data), we respectfully disagree that the Code Validation experiments are simple. The underlying APPS dataset [(Hendrycks et al., 2021)](https://arxiv.org/abs/2105.09938) contains coding problems ranging from “introductory” to “competition”. We do not agree that competition-level programming challenges are “simple”, nor that detecting subtle bugs in proposed solutions to such challenges is. Indeed, our results show that even a (near-)state-of-the-art model such as GPT-4o fails to do much better than chance on such a problem, indicating that it is in fact highly non-trivial. With that said, both our theory and our codebase support applications to many other domains, and we believe that this is a promising avenue for future research.
>
> ## Questions
>
> As requested by the reviewer, we have responded to their comments under the “Weaknesses” section. Following their suggestion, we have also added experimental results regarding zero-knowledge protocols and further discussion to our paper (described further in the update summary above).

---

> ### Author Response · Authors · 2024-11-25
> **Polite Reminder to Reviewer AhRe**
>
> Given that the discussion period is due to end in around 48 hours, we wish to offer a brief reminder of the changes we have made in response to the reviewer’s comments and the answers to their questions, as we have not yet received any reply to our response. In particular, we have:
>
> - Clarified the motivation behind, and importance of, zero-knowledge neural interactive proofs;
> - Clarified the significance of Stackelberg equilibria and the connection to bi-level optimisation;
> - Conducted additional experiments that connect more closely to our theoretical contributions;
> - Corrected the typo pointed out by the reviewer;
> - Explained how the code validation task is in fact non-trivial for state-of-the-art LLMs and thus why we do not believe the experiment is too “simple”.
>
> To the extent that these updates and replies address the reviewer’s concerns, we kindly request that they consider the impact this might have on their assigned scores; to the extent that they fail to address the reviewer’s concerns, we warmly welcome any further questions or comments, and will do our best to address them in the remaining time permitted.

---

> > ### Comment · Reviewer_AhRe · 2024-11-26
> >
> > Thanks for the response. I appreciate the clarification, but I am still not fully convinced on the usefulness of zero-knowledge in this setting. For example, in code validation, why do you expect the verifier and the prover need to satisfy zero-knowledge?
> >
> > For the Stackelberg Equilibria and bi-level optimisation, what I meant was: this is quite trivial of considering from the perspective of Stackelberg Equilibria and using bi-level optimisation in the end. In other words, I am not fully convinced the technical novelty.

---

> ### Author Response · Authors · 2024-11-27
> **Further Response to Reviewer AhRe**
>
> We thank the reviewer for their reply to our response, and we are glad that our explanations seem to have alleviated some of their concerns. We address the remaining concerns in their reply in turn.
>
> **Zero-Knowledge**
>
> First, as we highlight in our original response, we do not expect that the verifier and prover “_need_ to satisfy zero-knowledge” (emphasis ours), only that it may be _useful_ in some cases. Moreover, we would like to emphasise that our paper is not _specifically_ about zero-knowledge proofs. Indeed, the reviewer will note that they are described in Section 5.1, with the title of Section 5 being “Extensions”. We summarise what we view to be our core contributions further below.
>
> With that said, to try to explain the usefulness of zero-knowledge protocols further, let us consider three examples in which a user employs a weak (trusted) verifier agent to interact with a strong (untrusted) prover agent that is provided by some model provider via an API. The key idea of zero-knowledge protocols is for the prover (controlled by the model provider) to adopt a strategy that prevents the verifier (controlled by the user) from extracting additional knowledge from their interactions, while still being able to solve the task at hand.
>
> 1. First and foremost, recent work has highlighted the risk of “model stealing” via black-box queries [(Oliynyk et al., 2023)](https://dl.acm.org/doi/10.1145/3595292). In other words, merely by interacting with the prover, a verifier can distill the model representing the prover to gain more knowledge than is required to merely solve the task at hand.
> 2. Another concern is that the verifier might be able to query the prover in a way that allows them to develop dangerous capabilities ([Phuong et al., 2024](https://arxiv.org/abs/2403.13793); [Jones et al., 2024](https://arxiv.org/abs/2406.14595)). For example, a prover being queried about a chemistry problem might inadvertently leak information that enables the verifier to synthesise dangerous chemicals.
> 3. Finally, proprietary models serving as provers may have learnt private information (either deliberately via instruction from the model provider or unintentionally because of the information being in the pre-training dataset). It is important that we develop techniques to prevent the extraction of this data via black-box model access [(Carlini et al., 2020)](https://arxiv.org/abs/2012.07805).
>
> While we certainly do not claim to have directly solved these problems in our paper (as they are not the main focus of our work), we do suggest that the concept of zero-knowledge interactions between such agents provides a firm theoretical foundation for addressing such problems. On the other hand (from the verifier’s perspective instead of the prover’s), it also suggests a fundamental limit to the amount that might be learnt from interacting with another, more powerful agent.
>
> **Stackelberg Equilibria**
>
> We fully agree with the reviewer that using bi-level optimisation to solve for Stackelberg equilibrium is not a novel contribution of this paper, and indeed we do not claim it to be. In fact, only a mere 18 lines (3.3%) out of the 539 in the main body of our paper are dedicated to discussing this point, and we explicitly cite several prior works when explaining how these ideas are relevant to solving the problems we are interested in. Rather, the novel contributions of our paper are:
>
> 1. A unifying game-theoretic framework that generalises existing neural IP protocols;
> 2. Several new neural IP protocols, including those that allow for zero-knowledge proofs;
> 3. A theoretical and empirical comparison of both new and existing protocols;
> 4. An extensible codebase for testing different protocols in different domains.
>
> The reason that we briefly discuss solving for Stackelberg equilibria using multi-agent reinforcement learning is that some readers may not be as familiar as the reviewer with the standard techniques used for doing this, or how they can be tractably approximated using various different learning updates. We stress that we do not wish to claim this brief discussion as a novel contribution of our work, and we hope this alleviates the reviewer’s concerns.

---

> > ### Author Response · Authors · 2024-12-04
> > **Final Reply to Reviewer AhRe**
> >
> > As the discussion period draws to a close, we thank the reviewer again for their engagement. Due to the deadline, this will be our last chance to reply. We hope that even if the reviewer does not believe that every single issue with the paper has been resolved, that our responses further above have convinced them that, all things considered, our paper still makes sufficient contributions to meet the bar for acceptance. If so, we kindly ask them to consider updating their score.
> >
> > Specifically, we hope that our clarification regarding the contributions of our paper (and how solving for Stackelberg equilibria represents only a tiny portion of our work, which we do not claim to be novel) and the usefulness – at least in theory – of zero-knowledge interactions, will have alleviated the concerns of the reviewer. For a more detailed concrete example and explanation of the latter we also refer the reviewer to [our final reply to Reviewer 32Gh](https://openreview.net/forum?id=R2834dhBlo&noteId=ZR6l8R4uh3) above.
> >
> > Thanks once again!

---

### Official Review · Reviewer_ZrkB · 2024-11-03

**Soundness:** 4
**Presentation:** 3
**Contribution:** 4
**Rating:** 10
**Confidence:** 2

**Summary:**

This paper introduces "Neural Interactive Proofs" (NIPs), a framework for enabling a computationally bounded but trusted agent (verifier) to learn to interact with more powerful but untrusted agents (provers) to solve decision problems. The work bridges concepts from interactive proofs in complexity theory with modern machine learning approaches. The paper makes four main contributions:

1. A unifying game-theoretic framework based on prover-verifier games that generalizes existing neural interactive proof protocols
2. Several new protocols, including single-prover (NIP), multi-prover (MNIP), and zero-knowledge variants, each with theoretical guarantees about their properties and computational power
3. A theoretical and empirical comparison of both existing and new protocols
4. Empirical validation on two domains: a graph isomorphism task, and a code validation task using large language models

The work aims to create a foundation for future research on neural interactive proofs and their application in building safer AI systems by providing ways for weaker systems to verify the outputs of more powerful ones. The authors provide theoretical results establishing correspondences between their protocols and valid proof systems, as well as empirical evidence.

**Strengths:**

This paper stands out for bridging theoretical and practical contributions while maintaining high standards of rigor throughout, as well as for the generality of the neural interactive proofs framework.

1. Originality:
    - Novel unification of interactive proofs with neural networks, bridging complexity theory and machine learning
    - Creative adaptation of game-theoretic concepts to create learnable verification protocols, particularly use of Nash and Stackelberg equilibria and their relationship to proof validity
    - The careful treatment of approximate equilibria to handle practical ML settings
2. Quality:
    - Strong theoretical foundations with rigorous proofs of protocol properties
    - Documentation of hyperparameters, architecture choices, and training procedures
    - Release of reproducible codebase for future research
3. Clarity:
    - Excellent high-level exposition of the key ideas and their implications throughout the paper.
        - I want to call out in particular the description of practical applications of zero-knowledge proofs (L368-369: preventing model stealing via black-box access): this was enlightening.
4. Significance:
    - Addresses a crucial challenge in AI safety: verifying outputs of powerful AI systems when the verifier is computationally bounded but trusted, and the prover is powerful but potentially untrustworthy
    - Provides a theoretical framework that can be extended to many verification scenarios
    - Demonstrates the approach through experiments on both toy problems (graph isomorphism) and large language models (coding challenge)
    - Opens new research directions in combining interactive proofs with machine learning
    - Potential impact on development of safer AI systems

**Weaknesses:**

I believe there are two central weaknesses of this paper, beyond what is listed in section 7 (lack of use of advanced methods, evaluation only on two domains, lack of evaluation of all protocols):

The biggest weakness is that the scope, motivation, and impact of NIP are not made clear.  Perhaps this is just a lack of context on interactive proofs and game theory on my part, but I cannot tell whether the combination of game-theoretic equilibria with generalization-from-examples for specifying the verifier will be widely applicable and revolutionary, or turn out to be relatively niche in practice.

Some more specific points in an attempt to drive at my confusion:

- Line 107 says “Importantly, we assume that it is impractical or impossible to convert the task description into a specification amenable to standard formal verification tools.” This assumption seems wrong.  Insofar as we are capable of training a non-interactive verifier, and willing to trust that verifier, we can just use the verifier itself (however large it may be) as the formal specification.  Applying more-or-less standard formal verification techniques to theorem statements that contain a full neural net is considered, for example, in [1].  Instead, it seems the assumption to make here is that the task is complicated enough that any system which is powerful enough to (non-interactively) verify the solution is larger or more complicated than what we’d be willing to trust as part of our specification.
- Proposition 1 (L242—245, L770) is used to motivate the insufficiency of `adp`, but the example constructed in the appendix is contrived as far as I can tell.  Sure, it is possible to gerrymander the set of possible prover protocols and verifier protocols so that Stackelberg equilibria are insufficient, but this seems to me to be severely unrealistic.  In realistic settings, we expect that computationally unbounded verifiers always have a winning strategy of “ignore the provers and solve the problem”, and so sufficiency of `adp` or `nip` must be evaluated with regard to some reasonable model of computational limitations.  I did not find any justification in the paper for believing that the game used to prove Proposition 1 is a reasonable toy model of computationally limited provers.
- I would very much like to see an example of a problem where the interactivity permits significant compression of the specification / theorem statement (thereby making the specification easier to trust, as per the motivation of Example 1).  In the absence of this, the case for `nip` over `adp` is much weaker.  The graph isomorphism example is too weak: a verifier for graph isomorphism can be hand-coded to run in polynomial time.  In problems like graph non-isomorphism, where interactivity allows bypassing an exhaustive loop with random checking, we don’t expect significant compression in an interactivity-based theorem statement over a non-interactive theorem statement.  However, transformers cannot use loops, so this theorem statement compression is not relevant to NIP over IP.  Can NIP be fully factored into the classical compression that interactivity gives in IP (that of loops, as far as I’m aware) and the generalization-from-examples discussed on lines 266—269, or is there some interaction between the interactivity component and the neural / game-theory equilibrium component that would suggest that Example 1 is realistic rather than hyperbolic?

Additionally, perhaps relatedly, there is no exploration of how performance degrades as the complexity of verification tasks increases.

A secondary weakness is that, while a valiant attempt is made to present the theory rigorously, the notation is not fully explained and the constraints in one of the central definitions (5) are contradictory.  I call this weakness secondary because it is obvious that the ideas are sound and the definitions are repairable.

Definition 3 in general is under-defined:

- Line 173: ∆ is used without being defined.  What is it?
- Line 174—175: The notation $C(i) := \{c \in C : i \in N \}$ is being used non-standardly.  Strictly interpreting set-builder notation would give $C(i) := C$ or $C(i) := \prod_{i\in N} C$ (depending on whether we are taking a union or a disjoint union), but this is clearly not what is mean.  Perhaps you meant to write $C(i) := \{c \in C : c(i) = 2 \}$?  (Note that defining $N = \{1, \ldots, n\}$ is inconsistent with taking $2 = \{0, 1\}$, so for consistency you have to use 1 for false and 2 for true unless you want to change the definition of $N$ or call out $2$ as a different set.)
- Line 174: “When $\mu(c, t)$ is deterministic” how is non-determinism defined here?  What indication was there that $\mu$ could be non-deterministic?
- Line 176: What does $\sim$ in $i \in N’ \sim \mu(t, c)$ mean?
- Line 178: What is $\rho$?
- Line 178: Is the message chosen randomly per-channel, per-timestep, per channel per-timestep, or per-game?

Definition 5 is also confusing:

- Line 223—225: What is $m^T$?  Is it the same as $m_T$ in L140?  And I presume $1_S$ is the indicator function on membership in $S$?
- Line 223—225: The direction of the inequality seems wrong.  If we are trying to minimize loss, then we want $\sigma$ to have lower verifier loss (be better) when the expected match with the indicator function is higher.
- Line 228—230: Criterion 3 is inconsistent.  L219 says $\mu(c, 0) = \emptyset$ for all $c\in C$, but here we say that $\mu(0, c^\dagger)$ (which I assume is a typo for $\mu(c^\dagger, 0)$, as otherwise it does not type-check) is non-empty.  L220 is also contradicted: $c^\dagger = \{ p \}$ for $p \in N^p$ but $c^\dagger$ is assumed on L220 to equal $\{ v \}$ for some $v \in N^v$, which is only possible if $N^p \cap N^v \neq \emptyset$.  But even if this is the case, it would still contradict criterion 2, which asserts that no verifier can solve the problem, precluding the overlap of provers and verifiers.

[1] Jason Gross, Rajashree Agrawal, Thomas Kwa, Euan Ong, Chun Hei Yip, Alex Gibson, Soufiane Noubir, and Lawrence Chan. “Compact Proofs of Model Performance via Mechanistic Interpretability.” 2024. [arXiv:2406.11779](https://arxiv.org/abs/2406.11779).

**Questions:**

The main confusion I want addressed is that described above in the weaknesses section as the primary weakness.

Beyond that, some minor questions and comments:

Questions:

- On lines 141—142 the message sequence in the prover-verifier game is stochastic, but on line 165 the loss function on the product strategy space is deterministic without any access to a random seed.  How is this tension resolved?  Does the loss function fix a random seed ahead of time?
- L218: Should $\cup$ be $\sqcup$?  That is, are provers and verifiers required to be distinct?  Otherwise “as $p$ or $v$ respectively” on L219 seems a bit misleading without a caveat that we may sometimes have $p = v$.

Minor comments:

- On line 140, $m_T \in {\top, \bot}$, but on 143—146, 0 and 1 are used instead
- On lines 143—146, $\langle p, v\rangle(x) \in \{0, 1\}$, but on lines 153—155, $\langle p, v\rangle(x)$ is an entire sequence of messages ($\mathbf{m}$).
- Line 179: “assume” should be “say” or “declare” (or just remove “we assume”); it seems that this is not an assumption so much as a definition of what it means for play to terminate.
- Line 228—229: $\mu(0, c^\dagger)$ should be $\mu(c^\dagger, 0)$.
- L218—219 uses $N_p$ and $N_v$ while most other lines use $N^p$ and $N^v$.  I assume this is a typo.

---

> ### Author Response · Authors · 2024-11-21
> **Reply to Reviewer ZrkB (Part 1)**
>
> We thank the reviewer for their careful reading and commentary on our paper. In what follows, we address the weaknesses and questions raised by the reviewer, in light of our updates to the paper described in the official comment further above. In particular, we have now fixed all mathematical/notational typos, and added further detail (see also below) that we hope clarifies the scope, motivation, and impact of NIP.
>
> ## Weaknesses
>
> **“The scope, motivation, and impact of NIP are not made clear”**
>
> We concede that given the wide range of background material and length limitations, there is inevitably a reasonably large degree of context needed to easily interpret some aspects of the paper. With that said, we have tried to ensure that the key messages of the paper are digestible, and we hope that our answers below and updates to the paper have improved it in this regard. In what follows, we respond to the three specific points raised by the reviewer.
>
> - The reviewer is correct that in a sense the verifier itself represents the formal specification, though we would still defend the claim that (unfortunately!) it is not yet the case that large neural networks are amenable to analysis using “standard formal verification tools”.
>     - The concurrent work of [Gross et al. (2024)](https://arxiv.org/abs/2406.11779) is closely related to our paper in that they are also focused on using small, learnt components to verify the performance of larger models. While they are able to obtain stronger guarantees using white box access (whereas we only assume query access), their approach does not yet scale to the kinds of problems and models we consider, for example, in our second experiment. A direction or future work that we are especially excited about is the combination of neural interactive proofs with other tools such as white-box access or the ability to simulate a model.
>     - Finally, the reviewer is correct that our emphasis is on problems where “any system which is powerful enough to (non-interactively) verify the solution is larger or more complicated than what we’d be willing to trust as part of our specification”, and we found this rephrasing to be especially crisp and helpful. As a final additional subtlety, in practice we will be interested not in “any [sufficiently powerful] system” but in those powerful systems that we have available to us. I.e., perhaps there is nothing in theory to prevent the construction of a trusted, powerful agent to solve a task, but we do not in fact have such an agent or know how to construct it.
> - We agree with the reviewer that the construction of the counterexample in the proof of Proposition 1 is rather contrived (though this of course does not make it any less correct). We furthermore agree that the key idea here is about the computational limitations of the agents. We would argue, however, that our proof is an attempt to highlight the importance of realistic assumptions, rather than eschew them.
>     - Concretely, [Anil et al. (2021)](https://arxiv.org/abs/2108.12099) prove their correspondence result by limiting the verifier to be deterministic (which will not be a reasonable model of most AI systems, despite the fact that deep down they are essentially deterministic), and by assuming that perfect soundness is possible (which is also not realistic). Our claim in Proposition 1 is to show that relaxing these conditions means their correspondence no longer holds. This issue is averted (as shown in Theorem 1) by considering approximate equilibria and by not blurring the distinction between soundness and completeness when considering worst-case outcomes.
>     - Separately, both the theory of interactive proofs and our experimental results suggest that greater interactivity (i.e. NIP over ADP) leads to improved performance (see also our further reply on this point below).
>     - Finally, zooming back out to the broader point: in this work we model the computational limitations of agents by circumscribing their strategy spaces (as in Definition 5, but see also Figure 2b), though empirically we also consider the impact of limited access to training data in Figure 5. While not unreasonable, this is clearly not a perfect model for all purposes, as it does not capture, for example, the difficulty in searching over a larger strategy space. As we note in our introduction, we view our paper as falling between one extreme (modelling these limitations using Turing machines) and another (using different kinds of neural networks). We are excited about the possibility of future research that attempts to further bridge this divide in order to enable powerful theoretical results to improve the safety of real-world systems.

---

> ### Author Response · Authors · 2024-11-21
> **Reply to Reviewer ZrkB (Part 2)**
>
> - Regarding the reviewer’s final question, we are not entirely sure we have fully understood everything, but we attempt to provide further clarity regardless, and we welcome any additional questions or corrections to our potential misunderstandings.
>     - Firstly, we point out that in the code validation experiments, it is empirically the case that interactivity (i.e. using NIP instead of ADP) allows a verifier of the same computational capabilities to achieve higher accuracy, though we are unsure if the reviewer would view this example as sufficient. Nonetheless, it suggests that Example 1 is in fact realistic and not hyperbolic. From a theoretical perspective, while this takes us further from our domain of expertise as researchers, it is possible to combine probabilistically checkable proofs (PCPs) with IPs in order to obtain the benefits of both: requiring the verifier to only query small portions of the proof, while using interaction to handle more complex problems [(Kalai & Raz, 2008)](https://link.springer.com/chapter/10.1007/978-3-540-70583-3_44).
>     - Secondly, regarding the interplay between interactivity and generalisation from examples, one key intuition behind this that representing the game between prover(s) and verifier in ‘extensive-form’ (i.e. with multiple rounds of interaction) is that, assuming the verifier has perfect recall, their strategy can be encoded more compactly using a behavioural representation, meaning that effectively they only need to consider their local state when making a decision. In other words, the tree-like space of interactions introduces additional structure to the problem, factoring the verifier’s strategy space, which can in turn make learning easier.
>
> Finally, in response to the reviewer’s suggestion, we have now added analysis detailing how performance degrades as the complexity of verification tasks increases (see Figure 8).
>
> **“The notation is not fully explained and the constraints in one of the central definitions (5) are contradictory”**
>
> We are extremely grateful to the reviewer for pointing out some of the minor issues with the notation and definitions. We agree with the reviewer that despite this, “the ideas are sound and the definitions are repairable” and in our updated manuscript (described further in the update summary) we have now repaired all of the flaws pointed out by the reviewer.
>
> Regarding Definition 3:
>
> - Line 173: $\Delta(X)$ refers to the set of all distributions over a set X and is defined at the beginning of Section 2.
> - Line 174-175: We apologise for the typo. We meant $C(i) = \{ c \in C : i \in c\}$, where $c \subseteq N$. In other words, a channel $c \in C$ contains a set of players, and $C(i) \subseteq C$ denotes the set of all channels to which player $i$ belongs.
> - Line 174: Given the meaning of $\Delta(X)$ and the type signature $\mu : C \times \mathbb{N} \to \Delta(2^N)$ in Definition 3 we hope it is now clear that $\mu$ can be stochastic.
> - Line 176: We use the (standard) notation $\sim$ in $i \in N’ \sim \mu(t, c)$ to indicate that $N’$ is sampled from $\mu(t, c)$.
> - Line 178: $\rho$ denotes the distribution that governs the drawing of random messages, which occurs when the protocol does not select any agent to move – it has no other special characteristics or functions, we simply used the letter $\rho$ to represent “random”.
> - Line 178: A random message $m^0_{c,t}$ is sent to channel $c$ at timestep $t$ whenever we have $\varnothing \sim \mu(c,t)$ (i.e., the protocol does not select any agent to play in channel $c$ at timestep $t$). Thus, it is possible that a random message may be sent in one channel at a given timestep, but not in another. In other words, the distributions $\mu(c,t)$ over subsets of players needn’t be correlated between channels $c$ (or timesteps $t$, for that matter).

---

> ### Author Response · Authors · 2024-11-21
> **Reply to Reviewer ZrkB (Part 3)**
>
> Regarding Definition 5:
>
> - Line 223-225: We apologise for the typo. The reviewer is correct that it should have been written $m_T$, not $m^T$. It is indeed the same $m_T$ as in L140. We use $T$ to denote the index of the final timestep of a trajectory. Finally, the reviewer is also correct that $1_S$ is the indicator function on membership in $S$ (which is defined at the beginning of Section 2).
> - Line 223-225: We apologise for the typo. The inequality should indeed be the other way round (i.e., a higher chance of success corresponds to a lower loss).
> -  Line 228—230: We apologise for the typo. We should have written $\mu(c^\dagger, 1) = \{p\}$ instead of $\mu(0, c^\dagger) = \{p\}$. In other words, the game proceeds by an input $x$ being sampled at time 0, before being immediately terminated after a single message is sent by a prover p in the channel $c^\dagger$ at timestep 1. The reviewer is quite correct that in the PVG model we assume that $c^\dagger = \{v\}$ for some $v \in N^v$. Criterion 3 represents a counterfactual scenario: if it were the case that we instead had  $c^\dagger = \{p\}$ for some $p \in N^p$, then it would be the case that $\min_{\sigma^p} \Ell^v(\sigma^p, \sigma^{-p}_u) \approx l^v_\star$. This counterfactual scenario is consistent with the reviewer’s correct observation that there is no overlap between provers and verifiers. We have now highlighted explicitly that this criterion is a counterfactual.
>
> ## Questions
>
> **Questions**
>
> - In effect, yes, we can view the application of a deterministic loss function to stochastic strategies as being calculated given a random seed that is fixed ahead of time.
> - Yes, the union $N = N_p \cup N_v$ is indeed disjoint in line 218. We have updated the notation to $\sqcup$ to reflect this – thank you for the suggestion. In addition, as the reviewer notes in their “minor comments”, our use of subscripts here is inconsistent with our use of superscripts elsewhere, and we have also fixed this typo.
>
> **Minor comments**
>
> - Thank you for pointing out the typo regarding the inconsistent use of $\top$ ($\bot$) and 1 (0). We have now removed all earlier usages of the former and replaced them with the latter.
> - We apologise for this typo. The correct usage is that $\langle p, v \rangle(x)$ denotes an entire sequence of messages ($\mathbf{m}$). We have now updated Definition 1 to reflect this.
> - The reviewer is correct that the “assumption” in line 179 is really more of a requirement about the structure of the game. We have updated the wording to reflect this, as suggested.
> - We have now corrected the typo on line 228-229 (as also noted further above).
> - The reviewer is correct that the use of subscripts in  $N_p$ and $N_v$ (instead of superscripts) is indeed a typo, and one that we have now fixed.

---

> ### Comment · Reviewer_ZrkB · 2024-11-22
> **More questions**
>
> Thank you for the edits to the paper and for your detailed responses.  I haven't yet fully digested them, but I want to make sure there's as much time as possible before the end of the reviewing period to address the specific points that might raise my score, so I'm going to follow up with some additional questions (in the “The scope, motivation, and impact of NIP are not made clear” category) now, after having read the responses only once.
>
> (I will note in passing that the point "We would argue, however, that our proof [in the counterexample of Proposition 1] is an attempt to highlight the importance of realistic assumptions, rather than eschew them." is well-made, and also that I have not yet managed to digest
> > Secondly, regarding the interplay between interactivity and generalisation from examples, one key intuition behind this that representing the game between prover(s) and verifier in ‘extensive-form’ (i.e. with multiple rounds of interaction) is that, assuming the verifier has perfect recall, their strategy can be encoded more compactly using a behavioural representation, meaning that effectively they only need to consider their local state when making a decision. In other words, the tree-like space of interactions introduces additional structure to the problem, factoring the verifier’s strategy space, which can in turn make learning easier.
>
> )
>
> It seems to me that an interesting application of NIP requires the following properties:
> 1. We are trying to establish some property $\mathcal P$ (e.g., code is good) of some object $o$ (e.g., the code).
> 2. We have some property $\mathcal H$ (e.g., there is no bug on this line of code) of some other sort of object $p$ (e.g., one line of code), such that:
>     1. For a typical $o$, if $\mathcal H$ holds on all choices of $p$, then we would believe $\mathcal P(o)$ is true
>     2. For most (or all) $o$ for which $\mathcal P(o)$ is false, it is easy to chose a $p$ (perhaps at random) such that $\mathcal H(p)$ is false
>     3. It is easy to trust (the small model that performs) the choice of $p$ and the correlated check of $\mathcal H$
>     4. It is infeasible to check $\mathcal H$ across all choices of $p$.
>
> In the interactive proofs I've seen (which is alas not that many), the power comes from having a vast computational gap between "for all" ("holds on all choices" in 2.1) and "there exists" ("it is easy to chose a" in 2.2).  But this sort of computational gap seems to me to be of a different kind than what I'd expect to see in powerful LLMs.  What is an example of a task where we have the following three things?
> 1. there's a property $\mathcal H$ that is simultaneously hard enough to check that we'd need an LLM, but one that is easy enough to check that we expect we could trust the LLM to check it
> 2. spot-checks of this property are enough to flag most bad examples
> 3. exhaustive checking of this property is infeasible
>
> Note that the code checking example seems to fail at (2) and/or (3) -- I expect that either the small models will not be good enough at finding the suspicious places to inspect that can trust that they will be able to find all (or even most) suspicious blocks of code, or else the small models will be good enough that they can do a more-or-less complete exhaustive check for suspiciousness.
>
> Perhaps another way of saying this is that the power of IP seems to come from being able to hide information about how a choice was made, and it is not clear to me that this extends to NIP.
>
> I expect that if I become convinced otherwise (either by examples of NIP being expected to shine from the above sort of power, or by being given another frame on what makes IP powerful that suggests that the extension to NIP will benefit from the same power), I'd be happy to update my score to a 10; I expect that without this, I'll keep my score at an 8.
>
> I'll post a follow-up comment with some background on my understanding of IP / NIP that led to the above property list, which I hope may be useful if the above seems incorrect to you.  I'll also aim to respond to the other comments before the deadline.

---

> > ### Comment · Reviewer_ZrkB · 2024-11-22
> > **Some background on my understanding, hopefully useful if the description of the power of NIP / IP seems wrong**
> >
> > As I understand it (please correct me if this understanding seems wrong), interactive proofs can be described roughly as follows:
> > - Mathematical logic has a two-player game semantics where there is a "prover" P and "disprover" D, where a proof corresponds to a (provably) winning strategy for P.  The prover gets to take their turn on existential quantifiers ($\exists$), disjunctions ($\vee$), etc, while the disprover gets to take their turn on universal quantifiers ($\forall$), conjunctions ($\wedge$), etc.
> >     - For example, to prove $\forall x, \exists y, x + y = 0$, the disprover gets to go first, picking a number for $x$ (say 10), and then the prover gets to pick $y$ (they would pick -10 here).  The prover has a winning strategy of always picking $y = -x$.
> >     - In the graph non-isomorphism problem, say trying to prove that $H \not\cong G$, we can make a theorem statement with a sort-of silly use of transitivity: $\forall$ graphs $X$, $X \not\cong G$ or $X \not\cong H$.  The disprover (verifier) gets to pick an $X$ (they will pick some permutation of either $G$ or $H$), and then the prover gets to pick which clause to try to prove.  If the prover never picks a clause that the verifier knows is wrong, then this is moderately strong evidence that the graphs are not isomorphic.
> > - Interactive proofs function when we have a property $J$ of some collection of objects $o$ where, on most (or all) objects $o$ for which $J(o)$ does not hold, there is a cheap (polynomial time?) randomized winning strategy for the disprover.  If the verifier plays this strategy, then any win by any prover is strong evidence that $J$ holds on the particular object of interest.
> > - Since it is the disprover's turn on universal quantifiers, we should expect most interactive proofs to have verifiers whose key strategy is of the form "pick a random object for the universal quantifier".
> > - This analogy is not quite tight, because the game semantics for logic allows each player to see the other player's choices, while in IP there is hidden state.  It seems to me that examples in IP involve carefully massaging theorem statements so that we can leverage the hiding of information to avoid needing to play the non-hiding version of the game to completion.  (If there are examples where this does not hold, I'm quite interested!)
> > - Nevertheless, it currently seems to me that the power of IP comes from verifiers being able to make randomized choices that are nevertheless correlated with their hidden state, where being able to check "all possible choices" would be enough to get a non-interactive proof.  (In graph non-isomorphism, if there is a function from all graphs to a label "not a permutation of $H$" or "not a permutation of $G$" such that it is correct on all permutations of $G$ and $H$, then $G$ and $H$ must not be isomorphic.). (Please tell me if this seems wrong --  I'm pretty new to interactive proofs.)

---

> > ### Author Response · Authors · 2024-11-22
> > **Reply to Reviewer ZrkB's Further Questions**
> >
> > We thank the reviewer for their prompt and detailed reply. We greatly appreciate this constructive dialogue and the reviewer’s efforts in giving us time to respond before the rebuttal period concludes.
> >
> > We appreciate the crisp statement of proposed properties that the reviewer suggests an interesting application of NIP requires. While we believe this statement is almost right, there is a subtlety that elides the power of interactivity. In particular, the characterisation of property 2.a) is more appropriate for problems of a “flat” or “linear” nature, where objects $p$ are largely independent from one another. An example of such a problem would be checking whether a low-degree polynomial over a finite field is zero everywhere, which can be done with high probability via the Schwartz-Zippel Lemma. For such problems, randomisation but not interactivity is required. In these cases, we can employ a proof system with three message rounds – i.e. where the verifier asks a question, the prover responds, and the verifier makes a decision.
> >
> > The problems on which (N)IP shines, however, are more complex, branching problems, which are “tree-like”  or “exponential” in nature (as opposed to “flat” or “linear”). The advantage of multi-round protocols like NIP is the ability of the verifier to ask questions _dependent_ on the answers to previous questions. In other words, a single rollout in an NIP system explores a single (random) branch in a tree of questions and answers, and is analogous to the fact that the game semantics in mathematical logic provides a truth value for sentences of arbitrary quantifier depth. A related example of such a problem is the Quantified Boolean Formula (QBF) problem, which is PSPACE-complete, where recall that, famously, IP = PSPACE [(Shamir, 1992)](https://dl.acm.org/doi/10.1145/146585.146609). The verifier’s ability to respond to the prover’s choices in such questions is critical for their ability to succinctly check the validity of the formula, and compressing this multi-round interaction into a single round requires substantive cryptographic assumptions and/or the use of different (e.g. multi-prover) protocols.
> >
> > Finally, we return to more practical applications and the use of LLMs. Firstly, we note that an empirical fact about today’s state-of-the-art models is that while they are remarkably adept at challenging short-horizon tasks (“checking $\mathcal{H}(p)$”, i.e. an edge in the tree), they often struggle with problems that involve longer chains of reasoning and more complex problem structures (“checking $\mathcal{P}(o)$”, i.e. the whole tree). While it might be the case that scaling up today’s models eventually overcomes this problem, it may also be the case that the models that are sufficiently simple/small enough for us to be able to trust – such as by applying other verification methods, e.g. those proposed by [Gross et al. (2024)](https://arxiv.org/abs/2406.11779) – are still only capable enough to solve these smaller problems.
> >
> > Relating this to the code validation example, checking that a program conforms to a specification involves checking correctness at all levels. Indeed, software often has exactly the “tree-like” structure we described above. In order to establish that a property holds of an entire module, we might for example need to check the functions within that module, the loops and conditional statements with those functions, and finally the individual lines. To see that the module is buggy, it’s not enough to look at lines in isolation; one must see how they relate to the rest of the code at other levels of abstraction. A good strategy for the verifier in the code validation scenario might therefore be to randomly select some portion of the code to ask the prover about, then drill down on smaller and smaller parts randomly until they can check that a single line does what the prover claims it does. Indeed, looking at the transcripts for NIP (Appendix D.2.5) suggests that this is roughly the strategy adopted by the verifier.
> >
> > We hope that this explanation has addressed your concern, but please feel free to ask any follow up questions (or indeed randomly select a single paragraph to drill down on!).

---

> > > ### Comment · Reviewer_ZrkB · 2024-11-23
> > > **Drilling down on a particular paragraph (nonrandomly)**
> > >
> > > > A good strategy for the verifier in the code validation scenario might therefore be to randomly select some portion of the code to ask the prover about, then drill down on smaller and smaller parts randomly until they can check that a single line does what the prover claims it does.
> > >
> > > This is exactly the [kind of] strategy that I am concerned is not good enough to trust, because it fails 2(b).  For most buggy code, I expect that this sort of random choice will work poorly, especially as bugs get more subtle.
> > >
> > > Consider, e.g., some code with an off-by-one error on a single line.  In order to identify the bug, the verifier needs to be able to zero in on that line of code.  But if the prover behaves as if the off-by-one error is absent except when directly asked about that line (i.e., if it responds as if it were being asked about the correct code whenever it is generating text), then the verifier cannot get enough information to zero in on that line of code, unless it can detect the suspiciousness of the broken line all on its own.  But if it can do that, then it shouldn't really need the power of the prover to verify the correctness of the code, right?
> > >
> > > (For a more real-life example, consider https://github.com/agl/curve25519-donna/commit/2647eeba59fb628914c79ce691df794a8edc799f, where the code was broken on only 19 out of 2^255 inputs; if the prover seamlessly interpolated between answers on the full range (when asked big-picture questions) and answers on the restricted range (when asked questions about blocks close to the invalid ones), it seems likely that the verifier needs to give an essentially exhaustive inspection of the code, unless it knows which inputs in particular are buggy.)
> > >
> > > QBF in IP is interesting!  Is the arithmetization of the problem essential in https://dl.acm.org/doi/pdf/10.1145/146585.146609, or would it work just as well to have the verifier choose all the universal quantifiers and the prover give all the existential quantifiers?  (I guess another way of asking this question is: what are the standard techniques for taking hard-to-find failures and blowing them up so they can be found easily at random, and what would an analogous (trustworthy) technique look like for bug-finding or some similar LLM-esque problem?)

---

> > > > ### Author Response · Authors · 2024-11-23
> > > > **Exploring a random branch**
> > > >
> > > > We thank the reviewer for their timely reply and below attempt to address their remaining questions and concerns.
> > > >
> > > > In response to the concern about the described strategy not reliably working, we note that (N)IP protocols only require a _chance_ of finding the bug, which can then be improved by running the protocol multiple times. In fact, this is precisely what we see in practice if we compare Figures 3a and 3b. While the mean accuracy of NIP is around 65%, if we repeat the protocol ten times there are only about 10% of all inputs that the protocol misclassifies every time (and some of those will be false negatives, which represent a safe if overly conservative outcome). Clearly 10% is still too high for perfect security, but in this work our aim is to advance the state of the art rather than provide a flawless solution (desirable though that may be).
> > > >
> > > > At the same time, simply repeatedly spot-checking individual lines does not work in the same way because it leaves the verifier unable to exploit the tree-like structure of the problem. The latter strategy of course requires the verifier to use a more sophisticated reasoning process than mere random guessing, and in general we regard the extent to which contemporary ML models (such as LLMs) can learn this reasoning process to be one of the key open questions that motivates this work. Our results represent a tentatively affirmative answer in the domains we consider at least, but there is clearly much more work to be done in understanding the strengths and weaknesses of ML models in this regard.
> > > >
> > > > Regarding QBF, our (admittedly incomplete) understanding is that the arithmetization of the problem is essential in that it prevents the verifier from having to evaluate the entire resulting Boolean expression, and allows it to instead use succinct methods such as the sum-check protocol. Regarding the more general version of your question (and its relevance to LLMs), our understanding is that this is essentially a question of _amplification_. Classical techniques for amplification include the already mentioned repetition and randomisation of the protocol, but also reductions of the problem (as in the case of arithmetization for QBF) that allow it to be factored into smaller chunks from which errors propagate, and the perturbation of either inputs or the object itself (this could be purely random or adversarial). To the best of our knowledge, aside from repetition and randomisation (which we explicitly build into our approach), these amplification techniques are often quite specialised to the kind of problem being solved, though it is worth noting that multi-prover protocols can also be viewed as a form of amplification (i.e. the introduction of a second prover helps to amplify the errors of the first, and/or vice versa).
> > > >
> > > > In the case of ML models and LLMs, while it might be possible to pre-process problem instances using hand-coded techniques, one of our questions is whether such techniques can actually be (implicitly) _learnt_ by the models. Indeed, the dream would be to see whether ML models can either recover known non-trivial techniques that convert a problem representation into one that makes failures easier to find, or (more excitingly) to discover previously unknown amplification techniques. While we were not able to demonstrate this _directly_ in our current work (and we would need to do much more work even to understand how to design an experiment to test for this ability), we believe this is an especially interesting question for future work to examine. In the meantime, however, we note that even while we might not fully understand the internal mechanisms via which the models are representing and transforming the problem instances, it is empirically the case (as demonstrated _indirectly_ by our experiments) that they are capable of doing so in a way that allows them to outperform our non-protocol baselines.

---

> ### Comment · Reviewer_ZrkB · 2024-11-26
> **Amplification techniques**
>
> I think it's definitely worth adding a sentence or two to the discussion / future work (or even a whole appendix section, if there's time) about how essential amplification techniques are; it now seems to me that the two things we'd need to trust / prove about the smaller model / verifier are that it does a good job of checking, and that the amplification technique it is using is sufficient.  Of these, it seems like establishing that the amplification technique is sufficient to catch rare and subtle failures is going to be trickier by far, especially if we have to establish that it is possible to learn sufficiently good amplification techniques without necessarily being able to establish what the amplification technique is.
>
> In light of my this discussion, I will be increasing my score from 8 to 10 -- I now believe this paper lays solid groundwork for a substantial new field of investigation that may prove important to steering AI systems.
>
> However, I will be keeping my confidence level at 2 -- the existing literature in Interactive Proofs is still quite new to me, and I am not confident in my ability to judge how this paper fits in amongst existing work.
>
> I'd also like to thank the authors for their detailed and friendly engagement with my questions and concerns.

---

> > ### Author Response · Authors · 2024-11-27
> > **Reply to Reviewer ZrkB Regarding Amplification Techniques**
> >
> > We thank the reviewer for their continued engagement. We agree that the idea of amplification is highly important to the work we present in the paper, and this exchange has driven home the importance of mentioning that in the paper. While the deadline for making changes to the paper is rapidly approaching (in <24 hours) and so we may not be able to add this during the discussion period, we are already mid-way through creating an additional appendix section that surveys our paper’s connections to other related work in even more detail (including the [Gross et al. (2024)](https://arxiv.org/abs/2406.11779) paper), and this would be a good place to add discussion of the importance of amplification and other techniques for achieving it. We will make sure to add this appendix before the paper is published.
> >
> > We very much appreciate the reviewer updating their score in response to our changes and explanations. While we believe the reviewer might be being a little too modest regarding their confidence level (based on the quality of the questions they posed), we respect their choice :) Finally, we thank the reviewer for a helpful discussion – we believe the paper and our own thinking has benefited a lot from their comments and questions!

---

> > > ### Comment · Reviewer_32Gh · 2024-12-01
> > >
> > > (I'm another reviewer)
> > >
> > > I think it would be great if this paper included results on how much these oversight techniques improve if you're willing to spend more inference on them (e.g. by running the oversight procedure repeatedly and ensembling them). I think that this is an important question and I'm not aware of scalable oversight papers that explicitly measure it. (I know it's too late to update this submission, but maybe you should consider it for a future submission or as a post-acceptance change)

---

> ### Author Response · Authors · 2024-12-04
> **Quick Reply to Reviewer 32Gh**
>
> Thank you for this suggestion! In short, we completely agree. Amplification via repetition is one of the key ways in which classical interactive proofs gain their power and we are cautiously optimistic to see that this at least tentatively seems to be the case for neural interactive proofs as well, though of course this clearly demands a more detailed investigation than we were able to provide within the scope of the present work. Indeed, as the reviewer suggests, there may also be other ways in which inference-time compute can be used to boost the effectiveness of scalable oversight. We are in the process of analysing our results further in this regard and plan to conduct additional experiments along these lines. If we are satisfied that these further results are sufficiently robust we will at least include them in the appendix as a post-acceptance change (if, indeed, our paper is accepted!) and in any case will consider this further as part of ongoing follow-up work. Thanks again!

---

### Official Review · Reviewer_32Gh · 2024-11-04

**Soundness:** 3
**Presentation:** 1
**Contribution:** 2
**Rating:** 5
**Confidence:** 3

**Summary:**

This paper discusses various games which could be used as multi-agent training setups. The objective of these games is to come up with ways of training models by getting models to interact with each other; this is justified by reference to the problem of training models that are untrusted, and the problem of training models that are smarter than human labelers ("scalable oversight").

The paper provides some formalism for a variety of games that could be used for this purpose, proving various results about them (e.g. about cases where you can bound the worst-case risk).

It introduces some new protocols, such as the neural interactive proof protocol, and some zero-knowledge protocols.

It then evaluates some of these protocols in empirical settings; an algorithmic graph isomorphism setting and an APPS code verification setting.

**Strengths:**

- I think the problem discussed by this work is important
- I'm not aware of the NIP protocol being investigated in the past (though it is an obvious generalization of other things that have been studied). I think that the NIP protocol is plausibly better than other protocols that have been investigated, so it's a contribution to suggest it and empirically investigate it.

**Weaknesses:**

This paper has two main contributions: theory and empirical results. I'm concerned that the theoretical results aren't very important, they aren't crucially related to the empirical results, and the empirical results (while promising) aren't very detailed/strong.

## The theoretical results aren't very important

Firstly, I was not persuaded that a lot of the theory is very useful to understand. I assume that the results are correct, but I don't think they're very informative about how to construct better protocols or how to analyze the safety of protocols in this setting.

- I'm not sure that the proofs about worst-case performance can be used in practice, because the conditions are very strong.
- As a less important example, the paper introduces the possibility of zero-knowledge proofs, but only very briefly gestures at why it would be helpful to have a protocol be zero-knowledge in this context.

## The theoretical results aren't crucially related to the empirical results

I don't see a strong connection between the theoretical contributions and the empirical experiments. E.g. the zero-knowledge proof idea is proposed, very briefly justified, and then doesn't appear in the empirical results.

## The empirical results aren't very detailed/strong

The results on graph isomorphism are somewhat interesting but it's hard to take much away from them except as a demonstration that the nip protocol isn't buggy somehow.

The results on APPS have the potential to be really interesting--I can totally imagine a version of this paper where, along the lines of various scalable oversight papers in the past, you implement nip in the APPS setting and demonstrate that it works better than debate etc. But I don't feel like this paper establishes this conclusively or provides substantial insight into why the technique works (e.g. section 6.2 here is much less detailed than section 3 in Khan et al, "Debating with More Persuasive LLMs Leads to More Truthful Answers", https://arxiv.org/pdf/2402.06782).

----

I'm also not persuaded that prover verifier games are a good formalism for developing oversight techniques, compared to e.g. the type of formalism in "AI Control: Improving Safety Despite Intentional Subversion" https://arxiv.org/abs/2312.06942.

**Questions:**

- How is Figure 3, which shows (nip > adp > solo verifier > debate), consistent with Figure 8, which appears to show that at the end of training that (solo verifier = nip > adp > debate)?
- Can we get error bars in Figure 3, please?
- What future research directions are enabled by the theoretical contributions of this paper, and why are they important?

---

> ### Author Response · Authors · 2024-11-21
> **Reply to Reviewer 32Gh (Part 1)**
>
> We thank the reviewer for their careful reading and commentary on our paper. In what follows, we address the weaknesses and questions raised by the reviewer, in light of our updates to the paper described in the official comment further above. In particular, we have expanded our empirical results and linked them more closely to our theoretical contributions (the latter of which we also justify further below). To the extent that these replies address the reviewer’s concerns, we kindly request that they consider the impact this might have on their assigned scores; to the extent that they fail to address the reviewer’s concerns, we warmly welcome any further questions or comments.
>
> ## Weaknesses
>
> **“The theoretical results aren't very important”**
>
> While we understand that this may not have been that reviewer’s primary contention, we would defend the importance of theoretical results in “construct[ing] better protocols or … analyz[ing] the safety of protocols”. Indeed, many ideas in scalable oversight (including debate, for example), have been directly inspired by deep results in computational complexity theory that give us hope that a weak verifier may be able to oversee a strong prover. More than this, however, we believe that such results can be practically useful, informing us, for example that:
>
> - Optimising for Nash equilibrium is not sufficient for finding a valid protocol [(Anil et al., 2021)](https://arxiv.org/abs/2108.12099)
> - Several subtle aspects of the interaction protocol can have a significant impact on its resulting efficacy, such as:
>     - The ability of the verifier to randomise (Proposition 1)
>     - Separate communication channels for the provers (Theorem 2; see also [Barnes & Christiano, (2020)](https://www.alignmentforum.org/posts/Br4xDbYu4Frwrb64a/writeup-progress-on-ai-safety-via-debate-1))
>
> Regarding our results about worst-case performance, we agree that this setting is inherently intractable for extremely complex, real-world scenarios. Our aim with Proposition 2 is to gesture at the high-level conditions of a problem that imply that despite this difficulty it can be enough to minimise the empirical risk. As more advanced techniques and theory become available for targeting worst-case optimisation (see, e.g. [Shalev-Shwartz & Wexler (2016)](https://proceedings.mlr.press/v48/shalev-shwartzb16.pdf), [Tang et al. (2019)](https://arxiv.org/abs/1911.03618), [Gu et al. (2023)](https://openreview.net/forum?id=PvDY71zKsvP), etc.), satisfying these conditions may become available by other means. Our aim with Proposition 3 is merely to formalise the intuitive idea that the introduction of an adversary is a natural example of one such technique and mirrors, for instance, the use of an adversary in the debate protocol. To complement these theoretical results, we have also now included empirical results regarding the worst-case performance of different protocols (see Figures 3b and 4c), which indicate that progress can indeed be made in this direction.
>
> Regarding the motivation behind our discussion of zero-knowledge protocols, the idea is that while prover-verifier games may describe a training setup (in today’s current ML paradigm where there is a training-deployment dichotomy), in the future we will likely have large numbers of AI systems and services interacting with one another in order to solve tasks (see, e.g., [Drexler, (2019)](https://www.fhi.ox.ac.uk/wp-content/uploads/Reframing_Superintelligence_FHI-TR-2019-1.1-1.pdf) for one vision of this scenario). While we may want such systems to be able to query one another we may not wish for agents to gain additional knowledge from doing so (perhaps because it represents private information, or could imbue the agent with new, potentially dangerous capabilities). While this risk is not novel, the concept of zero-knowledge interactions between such agents provides a firm theoretical foundation for addressing such problems. On the other hand (from the verifier’s perspective instead of the prover’s), it also suggests a fundamental limit to the amount that might be learnt from interacting with another, more powerful agent.

---

> ### Author Response · Authors · 2024-11-21
> **Reply to Reviewer 32Gh (Part 2)**
>
> **“The theoretical results aren't crucially related to the empirical results”**
>
> We agree that this was a shortcoming of our original presentation, and have now included preliminary results for the zero-knowledge protocols in the graph isomorphism setting, validating our zero-knowledge training scheme (described further in the update summary). We have additionally implemented methods for adversarial training and algorithms for Stackelberg Policy Gradient [(Fiez et al., 2020)](https://proceedings.mlr.press/v119/fiez20a.html), Learning with Opponent-Learning Awareness [(Foerster et al., 2018)](https://arxiv.org/abs/1709.04326), and Stable Opponent Shaping [(Letcher et al., 2019)](https://arxiv.org/abs/1811.08469) in our codebase. Finally, in the new experiments that we include in the revised paper we have specifically sought to further study the worst-case performance, showing empirically how different protocols (see Figure 3b) and training regimes (see Figure 4c) affect this metric. As noted above, while optimising the true worst-case loss is intractable in practice, we argue both that this represents the gold standard for developing safe, advanced AI systems, and that progress can be made vis-a-vis this standard.
>
> **“The empirical results aren't very detailed/strong”**
>
> We agree with the reviewer that the point of the Graph Isomorphism experiment is as a validation tool. However, the highly structured and theoretically crisp nature of the resulting learning problem additionally allows us to validate the effects of different model sizes (Figure 2b) and amount of data (Figure 5) on the efficacy of the resulting protocol.
>
> We thank the reviewer for their compliments on the APPS setting. In response to their concerns we have added additional experiment analysis (described further in the update summary). In particular, further investigate the impact of problem difficulty, the distribution of protocol accuracies across problems, and the precision and recall of the different protocols. This helps point out a key reason that NIP and MNIP outperform debate which is that using standard expert iteration leads to the verifier rejecting very often under the debate setting. Our leading hypothesis is that this stems from the presence of a prover arguing for rejection, not just acceptance, combined with the fact that for code correctness there is a certain kind of asymmetry where arguing for the presence of a single flaw that undermines the correctness of a program is intuitively easier than arguing for correctness of the entire program. Another way to phrase this is that the debater arguing for rejection need only make an existential claim (“there is a line with a bug”), whereas the debater arguing for acceptance must make a universal claim (“all lines are bug-free”). Combined with the fact that solo verifier is also reluctant to accept any potentially buggy code (even after extensive prompt tuning to prevent this behaviour), this helps to explain the biassed and thus lower performance of debate relative to NIP and NMIP. When this bias is removed using stabilised experience replay, debate’s performance improves (see Figure 4b), though our ongoing and future work that tests additional learning algorithms and problem domains is likely to add further clarity here.
>
> **AI Control and Prover-Verifier Games**
>
> While we believe that the AI Control setting [(Greenblatt et al., 2023)](https://arxiv.org/abs/2312.06942) is an extremely important one and worthy of much greater attention, we would argue that this setting is importantly different from that of prover-verifier games (PVGs). The emphasis in AI Control is on detecting a malicious agent during pre-deployment testing. This has important implications that are both theoretical and practical. For example, the fact that the agent (the red team) is effectively in “simulation” means it must be modelled as memoryless. As an aside, the reviewer may be interested in concurrent work that formalises the AI Control setting [(Griffin et al., 2024)](https://arxiv.org/abs/2409.07985v1).
>
> While one can enforce such restrictions upon PVGs, they are a more general model, subsuming, for example, debate [(Irving et al., 2018)](https://arxiv.org/abs/1805.00899) as well as other protocols (see, e.g., [Hubinger (2020)](https://www.alignmentforum.org/posts/N64THGX7XNCqRtvPG/alignment-proposals-and-complexity-classes) for others). The emphasis in PVGs, however, is on training (or using) agents in a way that is robust to (potentially) partially misaligned agents, as opposed to being robust in the worst-case setting of a maximally misaligned, deceptive agent. In practice, these approaches are complementary: we might train an agent using a protocol (such as NIP) represented as a PVG before running pre-deployment tests as in AI Control, or we might want to test an agent using the AI Control methodology before using it with a PVG protocol (such as debate) once we know it is sufficiently safe to do so.

---

> ### Author Response · Authors · 2024-11-21
> **Reply to Reviewer 32Gh (Part 3)**
>
> **Presentation: 1**
>
> We note that the reviewer rated the presentation of the paper as “poor” (1), but apart from their remarks on missing error bars in Figure 3 (now added) and some lack of clarity regarding Figure 8 (now fixed, see below), they do not mention any other issues about presentation in their review. If there are other aspects of the presentation that the reviewer suggests we improve, we kindly ask them to communicate these so that we can do so; otherwise, we hope they will update their score regarding our paper’s presentation.
>
> ## Questions
>
> **How is Figure 3, which shows (nip > adp > solo verifier > debate), consistent with Figure 8, which appears to show that at the end of training that (solo verifier = nip > adp > debate)?**
>
> To clarify, Figure 8 was plotted so as to represent our training process, which, following [Kirchner et al. (2024)](https://arxiv.org/abs/2407.13692), uses Expert Iteration on partitions of the data at each iteration. We have now also included a plot of the test accuracy by iteration (Figure 12), which more closely aligns with the final test accuracy.
>
> **Can we get error bars in Figure 3, please?**
>
> Thank you for this suggestion. We have added error bars to the figure.
>
> **What future research directions are enabled by the theoretical contributions of this paper, and why are they important?**
>
> Thank you for the excellent question. While many of our theoretical contributions serve primarily to ground and motivate our subsequent discussion, we believe that the generalised PVG model we introduce and our preliminary theoretical results also provide fertile ground for future research directions, such as:
>
> - Comparing and creating new protocols using the generalised PVG formalism, which crisply captures the roles of features such as randomness (of agent strategies or of the protocol itself), hidden information (as in the use of separate communication channels), multiple agents (such as in debate or MNIP), and the solution concepts corresponding to valid outcomes of the protocols. While several protocols have been suggested at this point, it is far from clear that an “optimal” protocol has been discovered, and indeed a major motivation of our work is in comparing and creating protocols. By providing an explicit formalism we may also combine it with other formalisms, such as “open-source games” or “program games” (where at least one agent’s strategy is partially transparent to the other – in ML this could be enabled by providing white-box access to the model), and derive the guiding principles that we should employ if/when implementing new, more exotic protocols.
> - Developing and evaluating new algorithms for training agents in PVGs, such as those that aim to find different solution concepts (e.g. Stackelberg equilibria), or those place more emphasis on worst-case performance instead of average-case performance (e.g. by introducing additional adversaries). At the moment it remains an open question when it comes to the viability and impact of such algorithms, but our preliminary results (and codebase implementations) provide a foundation for additional investigations in this area.
> - Pursuing a theoretically principled investigation into the prevention of model extraction/stealing via black-box interactions (based on the idea of zero-knowledge interactions). While this direction is certainly more speculative than the others, we believe (as described further above) that the underlying problem is an important one, and one that may benefit from an alternative lens (namely, zero-knowledge interactions) when it comes to inspiring new solutions.
> - Developing a rigorous understanding of strategic interactions between agents of different computational abilities. In a future teeming with different AI agents, their capability asymmetries may be important not only for problems of scalable oversight but for problems relating to cooperation and exploitation [(Dafoe et al., 2019)](https://arxiv.org/abs/2012.08630). PVGs provide one example of settings in which a weak agent can solve a problem via interaction with a strong agent despite not being fully aligned with the strong agent, but understanding the more general conditions under which weak agents can be robust to exploitation by computationally stronger agents (and the corresponding “scaling laws”) is very much an open problem and is the subject of some of our research that builds on this paper.

---

> > ### Author Response · Authors · 2024-11-25
> > **Polite Reminder to Reviewer 32Gh**
> >
> > Given that the discussion period is due to end in around 48 hours, we wish to offer a brief reminder of the changes we have made in response to the reviewer’s comments and the answers to their questions, as we have not yet received any reply to our response. In particular, we have:
> >
> > - Explained the importance of our theoretical results;
> > - Conducted additional experiments that connect more closely to our theoretical contributions;
> > - Added further analysis and discussion regarding the results of our experiments;
> > - Clarified the differences between prover-verifier games and AI control;
> > - Further clarified our figures and added error bars (as requested by the reviewer);
> > - Answered the reviewer’s question on the research directions enabled by the theoretical contributions of our paper, and justified their importance.
> >
> > To the extent that these updates and replies address the reviewer’s concerns, we kindly request that they consider the impact this might have on their assigned scores; to the extent that they fail to address the reviewer’s concerns, we warmly welcome any further questions or comments, and will do our best to address them in the remaining time permitted.

---

> > > ### Comment · Reviewer_32Gh · 2024-11-29
> > >
> > > Thanks for your replies, and for these changes, and sorry for my late response.
> > >
> > > I continue to be unpersuaded by the conceptual arguments here. Here are some of my gripes:
> > >
> > > - I don't think your argument that zero-knowledge proofs are relevant is very strong.
> > > - I don't understand what you're saying about the relationship between this and AI Control. E.g. you say "The emphasis in AI Control is on detecting a malicious agent during pre-deployment testing", which is false.
> > >
> > > Overall I still don't understand concretely why the theoretical points you're making are important.
> > >
> > > ---
> > > Here are my updated thoughts on your empirical results.
> > >
> > > I think that you might be able to write a great paper about these protocols in the APPS setting, but I don't think the current paper lives up to that promise. Objections that I have include:
> > >
> > > - Figure 3c indicates that these techniques make extremely different precision-recall tradeoffs. So it's very hard for me to judge how good they would be at a consistent tradeoff: the results you've presented are consistent with any of the techniques except MAC being pareto optimal! It would be great if you were able to sweep over the precision-recall tradeoff for each technique and so allow an apples-to-apples comparison.
> > > - The differences between techniques are small and all of the techniques are quite bad as measured by accuracy.
> > >
> > > ----
> > >
> > > Here's an attempt at clarifying my complaint about presentation. My objection is mostly that the paper has many different sections that don't seem very connected, and it has many arguments mentioned in passing that I don't find very persuasive or concrete (e.g. the passing reference to why zero-knowledge proofs might be relevant).

---

> > > > ### Author Response · Authors · 2024-11-30
> > > > **Further Reply to Reviewer 32Gh's Latest Official Comment (Part 2)**
> > > >
> > > > We are not 100% unsure what the reviewer is referring to by “formalism” (as the original AI control paper is largely empirical and does not include an explicit formal model, hence the later work of [Griffin et al., (2024)](https://arxiv.org/abs/2409.07985v1)) but for the purposes of this reply we assume that they refer to the general setup of having an untrusted, more-capable model `U`, a trusted, less-capable model `T`, and limited access to trusted high-quality labor `H`, as well as the specific monitoring protocols proposed the paper. In this light, there are clearly some similarities between our generalised prover-verifier games (PVGs) and the protocols studied by [Greenblatt et al., (2023)](https://arxiv.org/abs/2312.06942). However, there are also some key differences:
> > > >
> > > > - Our formalism is more precise in that it _formally_ specifies exactly the structure of the interactions and the information available to agents (via different communication channels $C$ and the protocol $\mu$). As we noted in our previous reply, these technical subtleties can have important implications.
> > > > - It is also, in some ways, more general as it allows for adding other kinds of agents aside from the three listed above, though notably we _do not_ explicitly consider the use of human labour.
> > > > - On the other hand, we _do_ assume limited access to some ground-truth data, from which we hope to _train_ verifier agents to interact with the prover(s), whereas [Greenblatt et al., (2023)](https://arxiv.org/abs/2312.06942) focus on a number of handcrafted verification strategies that exemplify their approach (see Table 2 in their paper for a summary), in which the agents do not necessarily adapt over time.
> > > > - The emphasis in the original paper on AI Control is also slightly different insofar as it focuses on a broader methodology of conducting control evaluations of different protocols using a (human) red team and blue team. As noted before, we argue that this methodology is complementary to our work and could also be applied to the protocols that we discuss in our paper.
> > > >
> > > > We hope that this has helped to clarify the difference between the two papers, but please let us know if not.
> > > >
> > > > ### Empirical Results
> > > >
> > > > **Precision & Recall**
> > > >
> > > > We agree that the difference in precision and recall is highly notable, which is why we included Figure 3c. With a training scheme that allowed for weighting false positives and false negatives differently we would expect to see this balance change, though our current training scheme (iterated expert iteration) does not easily allow for this. In future work we plan to investigate additional training schemes to support this.
> > > >
> > > > On the other hand, we view this difference in trade-offs between protocols as a revealing empirical observation, which – to the best of our knowledge – has not previously been made in any other work on this topic (see, e.g., the concurrent related work cited in the following paragraph). While the reviewer is correct that it complicates the idea of selecting a “best” protocol, such selection can in practice be guided by the relative costs of false positives and false negatives (though false positives – e.g. letting through a piece of buggy code – will often be worse for the settings we consider).
> > > >
> > > > **Accuracy**
> > > >
> > > > Next, we agree that the differences between techniques are reasonably small. This is consistent with findings from concurrent related works such as [Kenton et al. (2024)](https://arxiv.org/abs/2407.04622) [see, e.g., Figure 2], [Khan et al. (2024)](https://arxiv.org/abs/2402.06782) [see, e.g., Figure 1], and [Arnesen et al. (2024)](https://www.arxiv.org/abs/2409.16636) [see, e.g., Figure 3], in which the variation in accuracy between different protocols seldom differs by much more than 10%. In these works, accuracy is often relatively low as well, consistently ranging between around 55-75% across different tasks. Given the challenging nature of the task we consider in our paper this is not especially surprising, though we agree with the reviewer that there is much room for improvement. We hypothesise that this overall lower performance in our case is due in part to the specific training algorithm we use (iterated expert iteration). In future work we plan to investigate additional training schemes.
> > > >
> > > > At the same time – in keeping with our theoretical formalism and moving beyond the works cited above – we note that part of the power of (neural) interactive proofs comes from being able to repeat them to obtain greater mean accuracy and lower worst-case accuracy. Figures 3b and 9 indicate that this is indeed possible in our setting (to a greater or lesser extent, depending on the protocol), which we take to be an encouraging sign.

---

> > > > ### Author Response · Authors · 2024-11-30
> > > > **Further Reply to Reviewer 32Gh's Latest Official Comment (Part 3)**
> > > >
> > > > ### Presentation
> > > >
> > > > We thank the reviewer for further clarifying their concerns about our paper’s presentation. Part of the challenge with this work is that our emphasis is on a unifying formalism and the comparison of several different methods, which means that with limited space the main body must necessarily lean somewhat towards breadth over depth (with depth covered in the appendices). While we have tried hard to maintain a good overall flow throughout the paper, we agree that this could be improved with additional signposting and some explanatory sentences here and there that serve to link the different ideas more tightly. Similarly, making slightly more use of our running example (which connects directly to the APPS experiments) and moving some of the further code validation experiments/analysis into the main body while moving some of the extensions and more speculative discussion to the appendices would be an extremely simple and quick change that would shift the balance of the main body towards the reviewer’s recommendation.
> > > >
> > > > Unfortunately, the deadline for updating the submitted PDF has now passed and so we cannot directly demonstrate such changes, but we sincerely hope that given the minor nature of these presentational changes compared to those that we have already readily implemented during the discussion period, the reviewer will trust us to implement these before the paper would be published. We appreciate the reviewer’s feedback on this matter and welcome any further suggestions.

---

> > > > > ### Comment · Reviewer_32Gh · 2024-12-01
> > > > >
> > > > > Thanks for your replies. And sorry if my messages here are missing some of your points.
> > > > >
> > > > > ## Re zero knowledge
> > > > >
> > > > > > To further clarify the reviewer’s concern, we wish to ask whether it is primarily about:
> > > > > >
> > > > > > The usefulness of having zero-knowledge protocols for model interactions at all;
> > > > > > Whether our proposed formalism is a reasonable way to characterise or learn zero-knowledge protocols for model interactions.
> > > > >
> > > > > My concern is mostly with the first.
> > > > >
> > > > > All the example applications are important problems. But I don't think that thinking about zero knowledge in the context of scalable oversight techniques is very relevant to helping with the applications you've named. For example, I don't see the practical connection between the zero-knowledge protocols in this paper and model-stealing. I would be persuaded if you could give an example of how a protocol you described in this setting would help substantially with any of the example applications you described.
> > > > >
> > > > > ---
> > > > > ## Re AI control
> > > > >
> > > > > > Our formalism is more precise in that it formally specifies exactly the structure of the interactions and the information available to agents (via different communication channels  and the protocol). As we noted in our previous reply, these technical subtleties can have important implications.
> > > > >
> > > > > I think that the structure of the interaction and the information available to agents is similarly specified in the original AI Control paper as in this paper: I think all the protocols in that paper spell out completely what information is included in each model call. So I still don't see an important difference here.
> > > > >
> > > > > From my perspective, the main difference between these papers is that the control evaluation methodology explicitly models some models as behaving adversarially with respect to the utility of the protocol designer, while this formalism assumes the models just myopically optimize reward (which tbc is also an interesting and important setting to study; I'm just not sold it's much of an advancement over any previous work).

---

> > > > > > ### Comment · Reviewer_32Gh · 2024-12-01
> > > > > >
> > > > > > ## Re empirical results
> > > > > >
> > > > > > Thanks for pointing out that previous work also has fairly similar accuracy results.
> > > > > >
> > > > > > > On the other hand, we view this difference in trade-offs between protocols as a revealing empirical observation, which – to the best of our knowledge – has not previously been made in any other work on this topic (see, e.g., the concurrent related work cited in the following paragraph). While the reviewer is correct that it complicates the idea of selecting a “best” protocol, such selection can in practice be guided by the relative costs of false positives and false negatives (though false positives – e.g. letting through a piece of buggy code – will often be worse for the settings we consider).
> > > > > >
> > > > > > I don't think that this is an interesting or robust result. In practice, if you wanted to change this tradeoff, you should probably do that by training the model to output a probability that the action was bad, and then applying some nonlinear function to that probability in order to do something with it. It would be very silly to change protocol just for the sake of changing the implicit threshold.

---

> > > > > > ### Comment · Reviewer_ZrkB · 2024-12-01
> > > > > > **Re zero knowledge**
> > > > > >
> > > > > > My understanding (the authors should correct me if their understanding differs) is that the application of zero-knowledge is that the labs and evaluators may have competing incentives, and zero-knowledge allows the labs to protect the model weights whole still allowing the evaluators to do their job.  If a lab is concerned that some evaluator may be trying to steal their weights (say, they want to provide an evaluation API for artifacts or services they create for companies), one of the strongest guarantees they could get, which might allow them to be more comfortable in deploying such an API, is that the evaluator could not clone the model for any task other than performing the evaluation.  I think in practice you'd need something stronger than the given ZK statement: you'd need not just existence of a stimulator (which is of course going to exist if you don't put computational constraints on it) but some sort of learning-theoretic bounds that say that seeing the messages on one example don't tell make it too much easier to learn the messages over other examples, even when the simulator is allowed to be comparably powerful to the prover.

---

> ### Author Response · Authors · 2024-11-30
> **Further Reply to Reviewer 32Gh's Latest Official Comment (Part 1)**
>
> We thank the reviewer for their time in continuing to engage and further spell out their concerns with the paper. We address their key points in turn.
>
> ### Conceptual Arguments
>
> **Zero-Knowledge**
>
> Regarding the relevance of zero-knowledge, we agree with the reviewer that this point is somewhat speculative. To further clarify the reviewer’s concern, we wish to ask whether it is primarily about:
>
> 1. The usefulness of having zero-knowledge protocols for model interactions at all;
> 2. Whether our proposed formalism is a reasonable way to characterise or learn zero-knowledge protocols for model interactions.
>
> If the reviewer’s concern is regarding the first point, we argue that being able to ensure zero-knowledge interactions would, if feasible, be useful. In particular, let us consider three toy examples in which an individual employs a weak (trusted) verifier agent to interact with a strong (untrusted) prover agent that is provided by some model provider via an API:
>
> - First and foremost, recent work has highlighted the risk of “model stealing” via black-box queries [(Oliynyk et al., 2023)](https://dl.acm.org/doi/10.1145/3595292). In other words, merely by interacting with the prover, a verifier can distill the model representing the prover (to gain more knowledge than is required to merely solve the task at hand).
> - Another concern is that the verifier might be able to query the prover in a way that allows them to develop dangerous capabilities ([Phuong et al., 2024](https://arxiv.org/abs/2403.13793); [Jones et al., 2024](https://arxiv.org/abs/2406.14595)). For example, a prover being queried about a chemistry problem might inadvertently leak information that enables the verifier to synthesise dangerous chemicals.
> - Finally, proprietary models serving as provers may have learnt private information (either deliberately via instruction from the model provider or unintentionally because the information is in the pre-training dataset). It is important that we develop techniques to prevent the extraction of this data via query access [(Carlini et al., 2020)](https://arxiv.org/abs/2012.07805).
>
> If the reviewer’s concern is regarding the second point, our justification for our proposed model is that it is (more or less) the closest possible analogue to the theoretical definition of zero-knowledge (as proved formally in Theorem 3). While it may be possible to approximate this and/or use a different model, we view this as a reasonable first step in establishing a principled theoretical foundation for future applied work.
>
> If the reviewer’s concern about the relevance of zero-knowledge is regarding neither the first nor the second point, we kindly welcome them to further clarify their concern so that we can better understand its nature.
>
> **AI Control**
>
> Regarding control, we apologise for any lack of clarity here. In our original reply, we were referring to control _evaluations_ specifically, which is listed by [Greenblatt et al., (2023)](https://arxiv.org/abs/2312.06942) as their primary contribution (“We describe a methodology to evaluate protocols that try to ensure safety despite intentional subversion”). We also note that as the third contribution listed in their paper, however, that they also introduce novel “[s]afety protocols and attack policies”.
>
> The reason we referred to control evaluations specifically was that we took this to be the “formalism” that the reviewer was referring to. The point of these evaluations (as with virtually all dangerous capability evaluations) is that they take place _before_ deployment (though this, of course, does not diminish the importance of additional run-time monitoring and oversight post-deployment). This is indeed evidenced by quotes from [Greenblatt et al., (2023)](https://arxiv.org/abs/2312.06942), e.g.:
>
> - “During the [control] evaluation, we aren’t trying to get the model to do actual useful work, and so the model doesn’t need to have access to dangerous affordances. So it should be possible to run the [control] evaluations in a much more secure and safe environment than the model will be run in during deployment.”
> - “Running the control evaluation needs to be much cheaper than running through all of deployment while still handling untrusted model strategies that involve rarely doing bad actions.”

---

> ### Author Response · Authors · 2024-12-04
> **Final Reply to Reviewer 32Gh (Part 1)**
>
> We thank the reviewer for their continued engagement as the discussion period draws to a close. Due to the deadline, this will be our last chance to reply. We hope that even if the reviewer does not believe that every single issue with the paper has been resolved, that our responses further above and below have convinced them that all things considered, our paper still makes sufficient contributions to meet the bar for acceptance. If so, we kindly ask them to consider updating their score.
>
> ### Zero Knowledge
>
> Thank you for the further clarification regarding your concern (and thanks also to Reviewer ZrkB for chiming in here!). We will try to better illustrate why we think the use of zero-knowledge protocols may be helpful in addressing the problems we identified by describing one of our examples in slightly more depth. First, recall that the key properties of a zero-knowledge interactive proof are as follows:
>
> - **Validity:** Roughly, a trusted verifier $v$ can interact with an untrusted prover $p$ in order to (with high probability) solve a specific task $T$. In the classical setting, $T$ is typically formalised as a decision problem, where we must classify instances $x$ as positive or negative, but this can be generalised to other kinds of problems.
> - **Zero-Knowledge:** For a given instance $x$, $v$ _only_ learns the classification of $x$ when interacting with $p$, not any other information. In the classical setting, this is (roughly) formalised by saying that there is no adversarial verifier $v’$ that can interact with $p$ in order to solve tasks beyond what they ($v’$) could solve with an oracle for $T$.
>
> Now, let us connect this to our examples. For the sake of illustration, we consider the second example, but similar analogies can be made in the other two cases. Suppose that we have the following:
>
> - $p$ is a powerful proprietary model developed by a major AI company to serve as a general purpose scientific assistant.
> - $v$ is a weaker personal software assistant used by an individual that can interact with $p$ via an API. (We also think that Reviewer ZrkB’s suggestion of an external auditing agent could be used here.)
> - $T$ involves classifying different chemical compounds based on complex features of their underlying structure. For example, $T$ could involve classifying chemicals based on what colour dye they will produce when combined with another chemical. (We apologise if this example is a little naive – we are not chemists – but we hope that the reviewer can at least imagine a similar kind of benign task.)
>
> Ideally, we would like $v$ to _only_ be able to perform task $T$ by interacting with $p$. We would not, for instance, like them to be able to repeatedly query $p$ in order to learn how to synthesise dangerous novel chemicals (assuming that simply being able to solve $T$ would not otherwise allow them to do this). In other words, we would like $p$ to adopt a strategy that makes the protocol for solving task $T$ _zero-knowledge_.
>
> A few remarks are in order:
>
> - First, we note that this is clearly a large oversimplification of the risk we are truly concerned with, and that the zero-knowledge property is somewhat conservative in this context (as there are presumably other kinds of benign information that we wouldn’t mind if $v’$ extracted). At the same time, we think it would be beneficial to have a principled, theoretical foundation for future work that at at least attempts (as the concept of zero-knowledge does) to answer questions such as:
>     - What does it mean for one model to gain knowledge from another?
>     - How would we measure this knowledge gain?
>     - What sorts of training regimes might, in principle, reduce this knowledge gain to its minimal level while still enabling the solving of a particular task?
> - Second, we note that the “threat model” here is slightly different to the main topic of study in our paper, where the risk is the untrusted prover misleading the trusted verifier, not the possibility of an _adversarial_ verifier extracting extra information from the prover (whether that prover is trustworthy or not). We apologise if this was unclear and will be sure to update our paper to mention this. Still, as pointed out by Reviewer ZrkB and as is hopefully made clear by our examples, we expect this threat model to be a significant one in the years to come, and one that can be neatly captured using prover-verifier games.
> - Third, as we emphasised in our original reply, we do not mean to suggest that the precise training setup we provided in order to learn zero-knowledge neural interactive proofs completely solves these problems. Rather, we hope this idea provides a novel lens on an important problem, and that it might help to inspire new solutions. This is consistent with Reviewer ZrkB’s remarks that our initial suggestions in this paper could be complemented in future work by learning-theoretic bounds, for example, as well as further practical exploration and experimentation.

---

> ### Author Response · Authors · 2024-12-04
> **Final Reply to Reviewer 32Gh (Part 2)**
>
> ### AI Control
>
> While we agree with the reviewer that the AI Control paper does spell out the nature of each interaction protocol, we do wish to stress that formally modelling such settings (as we do in our paper) can still be important. For example, [Griffin et al. (2024)](https://arxiv.org/abs/2409.07985v1) (the authors of which overlap with the original paper on AI control) formally model control evaluations as a partially observable stochastic game and use this model to introduce Pareto-improvements to the protocols originally studied by [Greenblatt et al. (2023)](https://arxiv.org/abs/2312.06942).
>
> We also agree with the reviewer that the main difference between our paper and the works on AI control is that “the control evaluation methodology explicitly models some models as behaving adversarially with respect to the utility of the protocol designer, while [our] formalism assumes the models just myopically optimize reward”, and that the latter is “an interesting and important setting to study”. We advance previous work by making the following contributions:
>
> 1. A unifying game-theoretic framework that generalises existing neural IP protocols;
> 2. Several new neural IP protocols, including those that allow for zero-knowledge proofs;
> 3. A theoretical and empirical comparison of both new and existing protocols;
> 4. An extensible codebase for testing different protocols in different domains.
>
> We would argue that the concurrent related works of [Kenton et al. (2024)](https://arxiv.org/abs/2407.04622), [Khan et al. (2024)](https://arxiv.org/abs/2402.06782), [Kirchner et al. (2024)](https://arxiv.org/abs/2407.13692), and [Arnesen et al. (2024)](https://www.arxiv.org/abs/2409.16636) complement but by no means obsolete or significantly diminish these contributions.
>
> ### Empirical Results
>
> We agree with the reviewer that a reasonable way to trade off precision and recall would be applying a nonlinear function to the original output of the protocol. Though to the extent that the precision and recall between different protocols differed so widely in our experiments and that – to the best of our knowledge – the breakdown into precision and recall has not been studied by other works that use these protocols, we still view it as worthwhile to bring this point to the attention of other researchers, as this implies that such subtleties might also apply to prior work. We hope that future work on various proof protocols will also watch out for this and take it into account more directly, as the reviewer suggests.
>
> ---
>
> Thanks once again!

---

### Author Response · Authors · 2024-11-21
**Summary of Responses and Updates to Paper**

We thank the reviewers for their careful reading and commentary on our paper. In particular, we appreciate their emphasis on the **novelty** (reviewers 32Gh, ZrkB, AhRe), **quality** (reviewer ZrkB), **clarity** (reviewer ZrkB), and **significance** (reviewers 32Gh and ZrkB) of our work. At the same time, we appreciate that our initial submission was not perfect. In particular we note that reviewers picked up on several key themes:

1. The completeness of our empirical results and their connection to our theoretical results (reviewer 32Gh);
2. The amount of analysis conducted using our experimental setup (reviewers 32Gh and AhRe);
3. Some minor typos and presentational issues (reviewers ZrkB and AhRe).

**In response to these initial reviews we have therefore introduced corresponding changes to our paper that we hope will help to address the reviewers’ remaining questions and concerns.** We add this comment here to provide a summary of those changes for transparency and to save ourselves from repetition in our responses to reviewers. To ease the assessment of these changes, we have highlighted new material in the updated manuscript in blue, though this will of course be reverted to black before publication. (Please note that due to space constraints, some of this new material is in the appendices.)

Finally, **we have also further explained the meaning and significance of our work and corrected some misunderstandings in our responses to individual reviewers.** Again, we hope that this has helped address the reviewers’ remaining concerns and confusions, and we warmly welcome any further questions and comments from the reviewers.

## 1. Additional Protocol Evaluations

- In the Code Validation experiment, we have now evaluated the missing protocols – MAC (“Merlin-Arthur Classifiers”, see [Wäldchen et al. (2022)](https://arxiv.org/abs/2206.00759)) and MNIP – and included them in our additional analysis (described further below). Overall, MNIP tends to perform best of all, whereas MAC performs slightly better than ADP, suggesting that both the use of multiple provers and also multiple rounds of communication can be beneficial (as implied by the underlying theory).
- In the Graph Isomorphism experiment, we have also implemented and tested the ZK-NIP protocol. (Unfortunately, the simulator losses in the ZK protocols require access to the agents’ logits on the model inputs instead of just the outputs, which are no longer available via the OpenAI API. We are in the process of introducing support for open-weight models into our codebase to allow for this in future work.) We observe that increasing the extent to which the prover prioritises the zero-knowledge condition does indeed lead to a tighter statistical zero-knowledge bounds, validating our training scheme, though this effect is modest.
- While not a separate protocol per se, we have also implemented and evaluated a new training method (‘stabilised’ expert iteration) that overcomes some of expert iteration’s shortcomings when used for learning in prover-verifier games. In this method the verifier’s guess is replaced with the true label in an annealed proportion of the rollouts, preventing a form of “mode collapse” in the precision and recall of the protocol.

## 2. Additional Experimental Analysis

- We have broken our analysis of the different protocols on the Code Validation task into precision and recall (see Figure 3c), helping to illuminate why some protocols perform worse than others, and motivating our introduction of stabilised expert iteration.
- Based on a suggestion from reviewer 32Gh, we have also examined the performance of NIP and the solo verifier on APPS problems of different difficulties. We observe a smooth trend (in Figure 8) with lower accuracy on harder problems, as we would expect.
- We further investigate not only the mean accuracy, but the worst-case performance of different protocols, complementing our theoretical discussions. The key result is shown in Figure 3b, where we see clear performance improvements using NIP, MNIP, and MAC, with smaller improvements for MAC and little improvement for debate. We also show how stabilising expert iteration can dramatically aid debate (in Figure 4c). A full set of histograms for accuracy distributions across the data is provided in Figures 9 and 10.

## 3. Fixed Typos and Improved Presentation

- We have now corrected the minor mathematical/notational typos identified by reviewer ZrkB, as well as the single natural language typo identified by reviewer AhRe.
- Definitions of worst-case robustness and worst-case uniform convergence had accidentally not been transferred from the paper’s body to Appendix B.2. This has now been amended.
- We have now added error bars to the plots, representing one standard deviation across 10 seeds.

---

### Meta-Review · Area_Chair_GUpH · 2024-12-21

**Metareview:**

This paper provides a framework that generalizes prover-verifier games where agents are parametrized by Neural Networks. They provide several protocols where the agents can leverage data and learn equilibria of the game to find valid proofs.

The strengths of this paper are:
- Conceptually, it aims at tackling an important problem
- It lays some strong theoretical foundations for prover-verifier games at the intersection between standard theoretical computer science, game theory and machine learning.
- Even though the paper has some significant theoretical contributions, the authors also provide some experiments showing the efficiency of the proposed protocol.

The weaknesses of this paper are the experiments and the potential lack of novelty with respect to the related work on debating agents.
After reading the paper and some of the related work, I believe that the conceptual (and theoretical) contributions are novel enough.


I suggest the authors to:
- Make their paper more accessible to the ML community, as the language used in the paper is not the most adapted to ML researchers. In particular, it would be helpful to:
- Be more precise regarding the usefulness of the theory and its takeaways. For instance, it could be useful to have a summary somewhere (e.g. in the contribution) of the theoretical insights (citing one of your answers here):
  - Optimising for Nash equilibrium is not sufficient for finding a valid protocol
  - Several subtle aspects of the interaction protocol can have a significant impact on its resulting efficacy, such as:
     - The ability of the verifier to randomize (Proposition 1)
     - Separate communication channels for the provers
- How these insights are implemented in practice in your methods (while not occurring in the baselines)
- Make a precise practical ablation regarding the importance of these theoretical insights (i.e. showing experiments that validate the theoretical claims above).

**Additional Comments On Reviewer Discussion:**

This submission led to long discussions between the authors and the three reviewers.  At the end of this discussion, Reviewer ZrkB increased their score to 10, but Reviewer 32Gh and Reviewer AhRe remained unconvinced, their main concern being that this work "is an obvious generalization of other things that have been studied.".

In the end, I believe that it was quite challenging to make a decision as the reviewers did not reach a consensus but I decided that the paper's contributions were novel and significant enough (which was IMO the main concerns of Reviewer 32Gh and Reviewer AhRe)

---

### Decision · Program_Chairs · 2025-01-22

Accept (Poster)